

# Towards the systematic reconnaissance of seismic signals from glaciers and ice sheets - Part B: Unsupervised learning for source process characterisation

Rebecca B. Latto[1], Ross J. Turner[1], Anya M. Reading[1], Sue Cook[2], Bernd Kulessa[3], and J. Paul Winberry[4]

[1]School of Natural Sciences (Physics), University of Tasmania, Private Bag 37, Hobart, 7001, Australia
[2]Australian Antarctic Program Partnership, Institute for Marine and Antarctic Studies, 20 Castray Esplanade, Battery Point TAS 7004, Australia
[3]Glaciology Group, College of Science, Swansea University, Singleton Park, Swansea SA2 8PP, UK
[4]Department of Geological Sciences, Central Washington University, Ellensburg, WA, USA

**Correspondence:** Rebecca B. Latto (beccablatto@gmail.com)

**Abstract.** Given the high number and diversity of events in a typical cryoseismic dataset, in particular those recorded on ice sheet margins, it is desirable to use a semi-automated method of grouping similar events for reconnaissance and ongoing analysis. We present a workflow for employing semi-unsupervised cluster analysis to inform investigations of the processes occurring in glaciers and ice sheets. In this demonstration study, we make use of a seismic event catalogue previously compiled
for the Whillans Ice Stream, for the 2010–2011 austral summer (outlined in companion paper, Latto et al., 2023). We address the challenges of seismic event analysis for a complex wavefield by clustering similar seismic events into groups using characteristic temporal, spectral, and polarization attributes of seismic time series with the k-means++ algorithm. This provides the basis for a reconnaissance analysis of a seismic wavefield that contains local events (from the ice stream) set in an ambient wavefield that itself contains a diversity of signals (mostly from the Ross Ice Shelf). As one result, we find that two clusters in-
clude stick-slip events that diverge in terms of length and initiation locality (i.e. Central Sticky Spot and/or the grounding line). We also identify a swarm of high frequency signals on January 16–17, 2011 that are potentially associated with a surface melt event from the Ross Ice Shelf. Used together with the event detection presented in the companion paper, the semi-automated workflow could readily generalize to other locations, and as a possible benchmark procedure, could enable the monitoring of remote glaciers over time and comparisons between locations.

## 1 Introduction

Cryoseismology is a field of glaciological research motivated by the possibility to infer aspects of glacier structure and processes from seismic signals, based on a continuous record of signals potentially from the entire volume of the glacier and its surroundings (VanWormer and Berg, 1973; Podolskiy and Walter, 2016; Aster and Winberry, 2017). The field has been enabled in recent decades by increased seismic deployments to remote locations (Kanao, 2018, Chapter 8). Networks of cryoseismic
instruments present the opportunity to monitor the non-tectonic seismic waves that are generated through a number of different



mechanisms in response to gravitational and thermal forcing, acting on bodies of ice. Investigating glaciers with such passive seismic methods adds to knowledge that might be generated based on data collected using in-situ and satellite methods (Wiens et al., 2008; Podolskiy et al., 2018). Seismic approaches can also inform the understanding of aseismic deformation through the detection of fine differences between viscous and elastic rheologies (Podolskiy et al., 2019). The range of cryoseismic

event types represents the diversity of mechanisms acting on and within a glacier. These include basal slip, melt water flow, and brittle failure (Nath and Vaughan, 2003). As a glacier moves intermittently over the bedrock and sediments beneath, it can experience basal stick-slip, resulting in signals at low frequencies of $10^{-2}$ Hz, lasting over 20–30 minutes (Pratt et al., 2014). Stick-slip is a cyclical process that first responds to built up elastic strain in sticky spots at the base and then releases that strain when lubricated by melt water (Winberry et al., 2011). Sticky spots usually arise on frozen ground where there is insufficient

water to decouple the glacier from the bed. Lubrication can come from englacial drainage pathways or regelation, which is defined as the melt response to high contact pressures.

Melt water flow can trigger a number of seismic responses from a glacier, examples include: water flow that results in fluid-induced resonance (Benn et al., 2009; Hammer et al., 2015); and changing water pressure resulting in crack opening or propagation (See Table S3 in Supplement; Colgan et al., 2016). As examples of such diverse signal types, the filling and drain-

ing of hydrological pathways (Fountain and Walder, 1998) produces relatively long, monochromatic, and harmonic seismic signals at frequencies between 1–10 Hz (Winberry et al., 2009a). In contrast, crevasse formation is a quick surface process (occurring within seconds) and generates seismic waves with frequencies between 10–100 Hz (Röösli et al., 2014). Surface fracturing can be triggered by increasing water pressure (Carmichael et al., 2012), contraction of the ice in response to changing air temperatures (Lombardi et al., 2019), tidal bending (Cole, 2020), or horizontal stretching (Podolskiy et al., 2016; Minowa

et al., 2019). Brittle deformation can also occur through the body of a glacier (Nath and Vaughan, 2003). Active processes at the downstream end of marine-terminating glaciers include basal outflow, iceberg calving and other ocean-ice interactions can also be studied using seismic events, although they are not a focus of this study.

Glacier-focused seismic deployments typically make use of arrays of multiple sensors, therefore, it can be a time-consuming process to analyze the data recorded over many months. Visual inspection by an experienced human analyst (manual event

detection) is highly informative, but not suited to large data volumes. It can also result in some aspects of the resulting event catalogue being analyst-dependent. A semi-automated approach is therefore desirable. The first step to automation, robust event detection, is addressed in Latto et al. (2023), which describes an algorithm designed for diverse, low signal-to-noise microseismicity from glaciers. The next step is event analysis, where data-driven approaches present an appealing way forward (Bergen and Beroza, 2019). We use the term 'event' broadly to include impulsive signals from glacier processes described

above, and waveform changes (such as an amplitude increase or frequency content change) with a less distinct onset that often arise in the seismic wavefield.

Machine learning is a general term for the application of computer algorithms to data for the purposes of prediction and pattern detection. It can be termed 'supervised' or 'unsupervised', where supervision in this context refers to the use of labelled data to train an algorithm (Marsland, 2015). The choice of system parameterization and algorithm is dependent on the presented

problem and the questions of interest. Supervised learning is often carried out using artificial neural networks to perform



classification and to build predictive models (Caruana and Niculescu-Mizil, 2006; Sibi et al., 2013). Other types of supervised learning algorithms for classification purposes include random forests and support vector machines (Cracknell and Reading, 2013). Unsupervised learning algorithms are 'inductive' as they develop the classification, or other model types, from properties of the data itself, i.e. not by using a labelled training dataset. Such approaches, ranging from minimally intensive to more

expensive, include k-means clustering, hierarchical methods, and self organizing maps (Jain, 2010).

The k-means clustering algorithm (k-means) is a relatively straightforward unsupervised learning algorithm, useful for partitioning high-dimensional datasets into groups of similar elements, called clusters (Anderberg, 2014; George, 2013). The k-means problem aims to converge on minimized Euclidean distance between each element to form the most well-separated clusters. Before solving, k-means requires the definition of two parameters: the number of clusters and the locations of the

cluster centers, called 'seeds'. One challenge is that the number of clusters representative of a dataset is not known a priori. Therefore, it is useful to compute the degree of separation between clusters in order to determine how similar the k-means solutions are for a preset number of clusters (Meilă, 2006; Zadeh and Ben-David, 2009). To determine the most advantageous locations for the cluster seeds, the k-means++ algorithm has been developed as a variation of the standard algorithm, whereby cluster seeds are uniformly dispersed at locations that are spread over the data such that any two seed locations are not too near

(Arthur and Vassilvitskii, 2006). Such a distribution improves k-means' speed and accuracy because the well-separated cluster seeds can better predict well-separated clusters. The k-means++ initialization procedure is more robust than a pseudo-random seeding and less prone to user bias than a deliberate choice of seed locations.

Machine learning applied to earthquake seismology is advancing rapidly due to the need for automated analysis given the wealth of available data (Bergen et al., 2019). Supervised learning approaches are able to draw on access to labelled data

(e.g. 1.2 million labelled seismic signal recordings; Mousavi et al., 2019). Unsupervised learning also benefits from labelled data in seismology because of the opportunity to validate and constrain results (Yoon et al., 2015; Galvis et al., 2017). Machine learning algorithms applied to seismic events often require as input the decomposition of events into attributes, called 'features', that typically describe the temporal, spectral, polarization, and network characteristics of a seismic time series (Riggelsen and Ohrnberger, 2014; Reynen and Audet, 2017). Discrimination between events is improved by careful selection of the features

used as input to the learning phase of algorithms (Mousavi et al., 2016).

The application of machine learning to geotechnical and environmental seismology is appealing because of the potential for the automated monitoring and classification of long, continuous, and relatively recent records (Hibert et al., 2019). Environmental seismology signals, as noted previously, can be varied in character, and difficult to distinguish from ambient (i.e. background) noise because of their weak amplitude. Further, labelled datasets are less commonly developed in environmental

studies. Therefore, unsupervised learning is a compelling approach for reconnaissance analysis of geotechnical and environmental seismicity records (Johnson et al., 2020). As examples, unsupervised learning has shown to be successful for classifying types of mine seismicity (e.g. k-means; Chamarczuk et al., 2020), discriminating between volcano-tectonic and rockfall events (e.g. self organizing maps; Köhler et al., 2010), and clustering landslide seismicity (e.g. deep convolutional neural networks; Seydoux et al., 2020). Such machine learning applications to lower energy signals especially benefit from the decomposition





of event catalogues to datasets of features because traditional discrimination between waveforms is difficult when all events are weak amplitude (Provost et al., 2017).

Applying machine learning to cryoseismology may enable discoveries regarding glacier dynamic and hydrological processes. Many cryoseismic signals are difficult to distinguish above background noise by the human analyst or traditional methods (e.g. Pomeroy et al., 2013). Unsupervised routines that can detect and classify events may distinguish some of the

underlying processes. In terms of event discrimination, machine learning can be used to differentiate between icequakes and earthquakes without prior knowledge of the structures of the cryoseismic signals (Jenkins et al., 2020). Similar applications can also detect calving events and avalanches in continuous seismic data (respectively, Köhler et al., 2012; Heck et al., 2018). Automated classification of cryoseismicity can enable investigation into processes which might otherwise be hidden from satellites or other in-situ observations. For example, automated detection and classification of seismic events has been used to

identify an ice-shelf fracture process induced by tidal bending, followed by resonance as seawater presumably fills the new fracture(Hammer et al., 2015). By studying the timing of icequake rupture, englacial meltwater flow has been related to other processes, such as heating by solar radiation and cooling induced by katabatic winds (Helmstetter et al., 2015a; Sawi et al., 2019).

The aforementioned studies demonstrate that as experience builds in applying machine learning to cryoseismology, there

is potential for different seasons and locations to be compared. A repeatable approach will afford the possibility to make comparisons between years for a given glacier environment, and compare different localities. The two companion papers presented as Latto et al. (2023), and herein, respectively demonstrate a systematic workflow in 1) building a catalogue and 2) using a simple clustering method as a reconnaissance tool to better understand glacier dynamic and hydrological processes.

We first review the seismic event catalogue compiled for the Whillans Ice Stream austral summer 2010–2011 deployment.

We then present and evaluate an unsupervised learning procedure, examining potential steps for constructing a dataset of informative features that capture all aspects of each seismic event while limiting biases. We implement the k-means++ clustering algorithm for grouping seismic events and explore k-means++ solutions supported by a manual appraisal of the event catalogue, investigating how the k-means++ solutions evolve, and choosing the logical number of clusters for the analysis. In Section 4, we interpret the clusters in terms of probable glacier processes and noise generation mechanisms that give rise to some of the

distinct groups of Whillans Ice Stream seismicity.

## 2   Whillans Ice Stream seismic event catalogue

The Whillans Ice Stream (WIS; previously known as Ice Stream B) is one of five major outlet glaciers of the Siple Coast (Fig. 1a) that together discharge 40% of the ice from West Antarctica to the Ross Ice Shelf (Price et al., 2001). The WIS is one of the fastest flowing ice streams in the Siple Coast, with velocities greater than $300\,\mathrm{ma}^{-1}$, due to its well-lubricated, deformable

subglacial bed (Tulaczyk et al., 2000). Negative mass balance (i.e. thinning of the ice column) is observed downstream (Bindschadler et al., 2005; Campbell et al., 2018). However, positive mass balance (i.e. thickening) is present in the upstream region of the WIS. This is because, in the middle of the 19th century, the neighboring Kamb Ice Stream (KIS) stagnated from 120





ma$^{-1}$ to a current flow speed of 10 ma$^{-1}$ (Retzlaff and Bentley, 1993; Joughin et al., 2002). The stagnated KIS, combined with thinning of the WIS, has resulted in a diversion of flow from the KIS to the WIS (Price et al., 2001). The result is thickening
in the accumulation area of the WIS, estimated at 1 ma$^{-1}$ (Bindschadler et al., 1993), which contributes to the deceleration of the WIS at an estimated rate of 5.5 ma$^{-2}$ (Beem et al., 2014). The deceleration, combined with bed strengthening that results from frozen ice at the glacier bed, is projected to cause a stoppage of ice flow in the next 40 years (Bougamont et al., 2003).

As a result of the dynamics of fast flow, subglacial hydrology, and basal mechanics, the WIS experiences a wide variety of cryoseismic events. The seismicity ranges from low frequency ($10^{-2}$ Hz) basal stick-slip (Winberry et al., 2009b) to possible
higher frequency (10–50 Hz) icequakes (Winberry et al., 2013). Glacier processes on the nearby Ross Ice Shelf exhibit seismicity with frequencies from $10^{-3}$ Hz ocean gravity waves and swells (Chen et al., 2019) to greater than 5 Hz elastic waves in the near-surface from temperature changes (Chaput et al., 2018). These processes from the Ross Ice Shelf could provide indirect seismic sources that can be detected in the external wavefield at the WIS (Wiens et al., 2016).

Continuous seismic recordings of the WIS were made between December 14, 2010 and January 31, 2011, inclusive (Win-
berry et al., 2010). Between the 35 stations deployed, two seismic sensor types were used: the Guralp CMG-40T-1 and Trillium 120 Broadband sensors. We use data sampled at 200 Hz from the sites with a Trillium 120 Broadband Sensor; 17 stations with names of format BBXX (Fig. 1b; triangles). Excluded are stations BB02, BB05, and BB09 due to missing components and/or incomplete data for a significant proportion of the deployment, resulting in 14 seismometers being used for this study.

Cryoseismic events occurring during the 2010–11 instrument deployment on the WIS were identified (Latto et al., 2023)
using an implementation of the multi-STA/LTA algorithm (Turner et al., 2021). Two catalogues are produced, a *reference event* catalogue and a *trace* catalogue, which list event information per seismometer. The reference event catalogue (1856 events) summarizes (e.g. averages) event information over the entire seismometer array and therefore summarizes the individual trace catalogue (8696 entries). The reference event arrival time can precede an individual seismometer's detected trace arrival time by half the network time for the three closest seismometers that detect an event. The reference event catalogue therefore maintains
the complete duration of an event across the network. Seismic events are preferentially detected across stations (Fig. 1b). For example, station BB07, which is located near to where the highest power seismic events nucleate during periods of local high tide, detected 1113 events; whereas station BB04, located further from the grounding line and tidal-related processes, detected 285 events. The unsupervised learning application described herein makes use of both the reference event and trace catalogues.

## 3 Unsupervised learning to identify glacier processes

We use an unsupervised learning approach applied to the previously generated catalogue of events (Latto et al., 2023). The machine learning workflow comprises the following steps: (1) Construct a dataset of 'features' (temporal, spectral, and polarization attributes of seismic time series) by processing the seismic records, guided by the pre-existing event catalogue; (2) Group the events into clusters using the k-means++ algorithm according to similarity between features identified; (3) Carry out exploratory data analysis, to review and optimize the clusters; (4) Identify probable glacier processes from the mechanisms





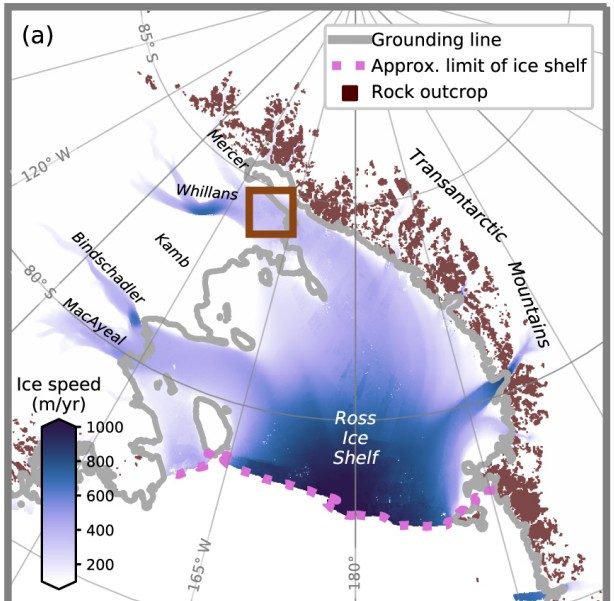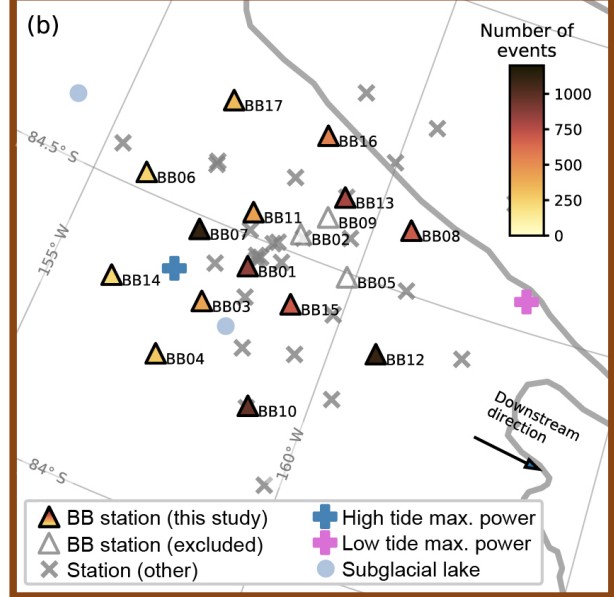

**Figure 1.** WIS dynamics and seismic environment. (a) Location of the WIS in the context of the Ross Ice Shelf (BB01: 84°17'43.8066"S 158°9'47.1636"W) with overlaid ice flow speed (Rignot et al., 2011), grounding line (grey line; Rignot et al., 2013; Mouginot et al., 2017) and rock outcrops (Burton-Johnson et al., 2016). (b) The WIS temporary broadband stations deployed in austral summer 2010 and 2011. The broadband stations in this study are colored according to the number of events detected per station, retrieved from the trace catalogue of Latto et al. (2023). The blue cross is approximately at the center of the stick-slip region, where the highest power seismic events nucleate during periods of local high tide (84.4°S 157°W; Pratt et al., 2014; Barcheck et al., 2018). The pink cross is at approximately the grounding line, where the highest power stick-slip events nucleate during periods of local low tide (84.55°S 163°W). The two pale blue circles indicate subglacial lakes (Wright and Siegert, 2012). Map generated by 'agrid' Python module (Stål and Reading, 2020).

that cause the more consistent clusters, through an appraisal of other recorded signals; (5) Interpret the origin of the seismic energy identified in the previous step.

### 3.1  Constructing a dataset of features

The representation of a seismic event by multiple features exploits all aspects of a waveform by the learning algorithm. We utilize a subset of features applied previously to a landslide seismic record (Provost et al., 2017) and refined for our application

to glacier processes (Table S1). We first construct a dataset of features from the waveform data, guided by the metadata in the trace catalogue (included as TraceFeatureDatasetWhillans.csv in Electronic Supplement). Then, we use the trace dataset of features to compute a reference dataset of features that correspond to the reference catalogue (included as ReferenceFeature-DatasetWhillans.csv in Electronic Supplement). Each of the 1856 reference events is thus characterized by a single feature set (as an appropriate average or median) that is representative of the recording stations. Feature-specific computation details are

given in Table S1. From this point onward, when referring to event features we are describing those pertaining to the reference feature dataset.





The set of features for each event are transformed prior to clustering to reduce potential biases. In this context, bias refers to an inconsistent weighting of the importance of a given feature based on numerically large ranges that can unduly influence clustering results because k-means relies on geometric (Euclidean) distances (Aksoy and Haralick, 2001; Mohamad and Usman, 2013). First, we review the distribution of each feature to determine a scaling (Table S1). For example, features that quantify energy typically fit exponential distributions and so will transform to a less biased distribution when set on a logarithmic scale. Other features that have very narrow ranges (e.g. the rectilinearity feature ranges from 0 to 1) can incorrectly skew towards values near their minimum when set on a logarthmic scale, so are kept on a linear scale. We carry out a normalization procedure to constrain the features to the same range, centered at zero, by scaling by a feature's standard deviation. After scaling and normalizing, the features demonstrate comparable distributions (histograms shown in Fig. 2; box-and-whiskers shown in Fig. S1).

### 3.1.1 Selection

The next step in our work flow is feature selection, which is applied here to remove features that can contribute to bias. The inclusion of too many correlated features in a clustering procedure can yield an artificial weighting. We compute a comparison of each feature pair, quantifying similarity by r, the Pearson correlation coefficient (Fig. 3). The maximum anti-correlation is r=-0.75 and the maximum (non-diagonal) correlation is r=0.99. We define similar features at r⩾0.95, after a careful review of the spread (variance) observed in feature relationships below that threshold (e.g. Fig. 3, scatter plot between feature 2 and feature 5 with an r=-0.75). Within each highly correlated set of features, we choose which feature is most useful to keep for analysis (scatter plots shown in Fig. S2; reasoning for feature choice provided in Table S2). After removal of a total of 7 features, the input to clustering (our 'standard' dataset) contains 30 remaining features to characterize each event.

### 3.2 Grouping the events into clusters

The dataset of selected features is used to find clusters of similar WIS events, by means of the k-means++ algorithm (Jain, 2010), which employs a standard initialization approach, with the default parameters for improving convergence as defined by the widely used Python implementation (Pedregosa et al., 2011, sklearn.cluster.KMeans). The data matrix used as input to k-means++ is defined as: $X = x_{ij}$, where $i = 1, \ldots, m$ and $j = 1, \ldots, N$, with $m$ waveforms described by $N$ features. The algorithm takes the following steps:

1. Initialize $k$ clusters, where each cluster $r$ is assigned a seed (centroid) $\boldsymbol{\mu}_r$ from $r = 1, ..., k$, i.e. each $r$ is defined by $\boldsymbol{\mu}_r = [\mu_{r1}, \mu_{r2}, ...\mu_{rj}, ...\mu_{rN}]$.

2. Assign waveforms $x_{ij}$ to a cluster by minimizing Euclidean distances between all the features of a waveform and a centroid $\mu$. The `argmin` function finds the value of r which minimizes the Euclidean distance. Each waveform $i$ is







**Figure 2.** Distribution of each feature in the feature dataset after log or linear scaling and normalization. The feature tags (top left corner of each plot) and color attributions (top to bottom: pink, green, blue, and purple) correspond to the numerical tag per feature and feature types provided in Table S1. Other qualities of each feature, such as the median, minimum, and maximum of each distribution, are provided in a box-and-whisker plot with the same number and color (Fig. S1).

assigned to a cluster number $r$ as represented by the following expression:

$$c_i = \operatorname{argmin}_r \left( \sum_j^N (x_{ij} - \mu_{rj})^2 \right).$$





**Figure 3.** Pearson correlation coefficient (r) matrix to quantify feature similarity from -1 to 1. The maximum anti-correlation of this dataset (r=-0.75) shows that features at this r value display high variance, shown by the scatter plot of an example pair of features, 2 and 5. At r=0, there is no correlation, exemplified by another pair of features, 2 and 4. The highest correlation of this dataset is at r=0.99, and the strong relationship of features at this correlation coefficient is demonstrated by a third example pair of features, 2 and 3. In this case, feature 2 is removed from analysis. The scatter diagrams of other features with high r values are provided with details on feature choices (Fig. S2; Table S2).

3. Recompute the centroid of each cluster based on the mean of the waveform features assigned to each cluster. Each cluster $r$ is assigned a centroid $\boldsymbol{\mu}_r$ defined by:


$$\mu_{rj} = \frac{\sum_{i=1}^{m} \delta(c_i, r) x_{ij}}{\sum_{i=1}^{m} \delta(c_i, r)},$$





where $\delta$ is the Kronecker delta function.

4. Iterate Steps 2 and 3 until a threshold indicating convergence is reached, i.e. when the centroids no longer significantly update.

## 3.3 Exploratory data analysis for optimization

Our implementation of k-means++ requires the number of clusters ($k$) to be prescribed a priori. Many studies have attempted to appraise the value for $k$ via statistical techniques (e.g. Legany et al., 2006; Bhargavi and Gowda, 2015). As an example, the Silhouette test quantifies the tightness of a cluster by computing the Euclidean distance of each n-dimensional data point to each cluster set of means (Rousseeuw, 1987). This test is, however, less informative when applied to real, noisy data whose clusters are not necessarily well-separated (see Sect. S3.3; Famili et al., 2004; He and Yu, 2019). Therefore, a recommended

approach combines user-domain knowledge, which qualifies the usefulness of a clustering result, with statistical metrics, which quantify how well-separated a clustering result is (Zu Eissen and Wißbrock, 2003). We employ a semi-automated methodology that complements the unsupervised k-means++ algorithm with an initial manual appraisal of the seismic event database and thorough review of parameter options. We aim to optimize the parameter choice of $k$ for this application to find clusters of cryoseismic events that can be reasonably matched to known glacier dynamic processes and/or used to investigate new or

previously unidentified glacier signals.

### 3.3.1 Evaluating the WIS event catalogue by human analyst

The manual appraisal is a thorough search and evaluation of the WIS catalogue by an analyst using a GUI tool designed for this application (Fig. S4). The tool provides fields for recording visually discernible features of an event, such as spectral description and maximum characteristic frequency, emergent behavior, and envelope description. The tool also provides a field

to assign each event a probable cluster type, thereby producing an ad-hoc labelled dataset of cryoseismic events for the WIS. The labels assigned during the manual search include processes such as 'stick-slip' and 'teleseismic' and broader potential attributions to 'Other' that is a catch-all category of noise-type events potentially generated from the surrounding Ross Ice Shelf and Ross Sea.

While the manual appraisal is conducted on all events, we have previously assigned low and high confidence labels to

each event in order to assess relative uncertainty in the dataset (Latto et al., 2023), with 35% of events being assigned a high confidence rating. The following cluster analysis and discussion uses all events, justified by the low and high confidence events presenting with logical patterns in the feature spaces and spatio-temporal domains (see Fig. 4). 'High confidence events only' figures are included in the Supplement for a subset of figures (Fig. 4, 5, 6, 7, and 8 with corresponding Fig. S5, S6, S9, S10 and S11).

We verify the manually appraised labels for events where the data are available. First, by comparing the Pratt et al. (2014, Supporting Information: Table 4) stick-slip catalogue with the manual appraisal labelled stick-slip events, we find 136 of our identified stick-slip events are identified as such by Pratt et al. (2014) and 4 are newly identified. In this case, by the word



'event' we are referring to any segment of stick-slip rupture, where the expected WIS stick-slip episode has two to three ruptures in a 30 minute span. In later discussions, we refer to the 136 verified stick-slip events as PRA14 (i.e. Pratt et al., 2014)

and the 4 additional events as PRA14 additional. The global seismic catalogue (U.S. Geological Survey, 2022) is also used as a cross-reference list with manually appraised teleseisms and other events not necessarily identifiable by eye as teleseisms. We find 68 events are potential teleseisms, which we label in subdivisions as Teleseism I ($> 3.5$ log a.u. peak amplitude, 32 events) and Teleseism II ($\leq 3.5$ peak amplitude, 36 events).

We summarize the results of the manual appraisal for the event catalogue in the bivariate feature relationships of duration,

peak amplitude, and characteristic frequency with analyst-appraised labels (Fig. 4). The representation of events in this two-dimensional framework illustrates how clusters are inherently formed by feature relationships. For example, the finding that all events labelled as stick-slip motion have similar durations ($10^2$–$10^4$ s), characteristic frequencies ($10^{-2}$–$10^{-1}$ Hz), and peak amplitudes ($10^4$–$10^5$ a.u.), lends credibility to the potentially mechanistic significance of clustering results. Variations within the features with assigned cluster labels suggest where a different number of clusters and corresponding labels might better

suit the feature dataset. Fig. S5 (with high confidence events only) compared with Fig. 4 shows that high confidence events are typically the longer and stronger events. The overall patterns inherent in the clusters, particularly for the stick-slip events, largely remain the same.

We aim to evaluate the labels assigned to the cryoseismic event types we expect from the WIS using overviews such as Ekström et al. (2003) and Podolskiy and Walter (2016) that document the common feature ranges of cryoseismic events

(Table S3). In doing so, we can improve the manual appraisal by, for example, considering distinct labels for some of the broad 'Other' events, such as 'ocean primary and secondary microseisms', for durations between 1–10 s and characteristic frequencies $10^{-1}$–1 Hz.

### 3.3.2 Investigating cluster evolution

By definition, the points in the clusters yielded by k-means++ with cluster parameter $k$ will rearrange when the cluster param-

eter is increased to $k + 1$ or decreased to $k - 1$. We are motivated to understand how the seismic events in a cluster split or group as $k$ changes in order to most reasonably determine the choice of $k$, the number of clusters. Typically, studying cluster evolution is challenging due to the built-in pseudo-randomness of k-means that assigns clusters a potentially different nondescript numerical label each time the k-means problem is solved. Therefore, instead of relying on the automated k-means label, we use the manual appraisal in combination with the number of seismic events assigned to each cluster, to more easily identify

the clusters as they appear and change for each $k$.

We can then apply the k-means++ algorithm to the standard feature dataset (Sect. 3.1.1) for number of clusters $k$=2 (the minimum possible) up to $k$=14 (a realistic maximum), and evaluate the corresponding solution as it evolves (Fig. 5). The illustration of how clusters merge and split as $k$ is increased offers two main conclusions. Firstly, as $k$ increases, typically only one of a set of clusters will split into two clusters, as depicted by the single assigned arrow per row that illustrates where a

portion of the seismic events of a cluster will form a separate group. This indicates that if a single cluster appears to represent two distinct types of events, that cluster could be pushed to split when clustering is forced to a solution at $k + 1$. Secondly, once





**Figure 4.** Manual appraisal of the reference event catalogue. The features measuring duration (log, seconds) and peak amplitude (log, a.u.) are retrieved from the reference event catalogue, and characteristic frequency (log, Hz) is determined from the spectrograms of each event in the manual appraisal. Event types that are identified from the manual appraisal and further validated in subsequent investigation show the natural groupings, or clusters, of events in each bivariate feature pair. The manual appraisal of the reference event catalogue showing high confidence events only is included in Fig. S5.

a larger cluster splits into a new cluster, that new cluster will generally retain its composition as $k$ increases. For example, the cluster in the second column from the right (i.e. first composed of 7.38% of the total events) appears to contain approximately the same number and composition of Teleseism I events as $k$ is increased (Fig. 5). An exception to both findings is shown in the third column from the left (i.e. first composed of 13.63% of the total events), which is formed by the split part of two




clusters, that immediately to the left and the furthest column to the right. This third column cluster splits at $k$=9. In routine use of unsupervised learning, detailed investigation of the cluster splits for sequential values of the k-means++ algorithm would not be expected; however, in this contribution we show cluster evolution to better inform the choice of $k$.

In the Supplementary Materials Sect. S3, we describe the quantitative approaches used to recommend an optimized choice of $k$=10 (Fig. S3). After contextualizing that choice with the previous appraisal of the catalogue and included discussion on cluster evolution, we choose to proceed with $k$=10.





**Figure 5.** Illustration of how clusters (depicted as circles) evolve in composition from $k=2$ (top row) to $k=14$ (bottom row). As $k$ increases (top to bottom), each column tracks an individual cluster and arrows indicate when a cluster splits or merges. The heavy grey line at $k=10$ indicates the preferred value of $k$, along this row, cluster numbers are counted from left to right (1 to 10) and match those in Table 1 and Figures 6 and 7. The number above each cluster is the % of total events, thereby the numbers across a row sums to 100%. Pie chart segments represent the percentage of events within a cluster, as labelled with an event type during the manual appraisal. Clusters 3, 4, 6, 7, 8, 10 contain events best characterized as noise-types based on the majority labelled as 'Other' and their features (Sect 3). Clusters 1, 2, 5, and 9 contain either labelled events and/or noise that appears related to processes by further analysis. Event types as identified by manual appraisal are shown in the legend; further event types identified by unsupervised learning (Fig. 6) occur in Cluster 5 (events from an icequake swarm, lasting 2 days). The illustration of how clusters evolve in composition from $k=2$ to $k=14$ showing high confidence events only is included in Fig. S6.



### 3.4 Event clusters and glacier processes

We evaluate the clustering solution for $k$=10 in order to infer the glacier processes potentially represented for each cluster (Table 1). For ease of referencing, the clusters yielded from $k$=10 are numbered (1–10 in the order shown in Fig. 5). For each cluster, we first make use of the manual appraisal to compute the percentage of events that are attributed to stick-slip processes and teleseismic earthquakes. Cluster 1 and Cluster 2 are assigned the majority of stick-slip events, Cluster 2 is also assigned a minority of Teleseism I events, and Cluster 9 is assigned the majority Teleseism I and Teleseism II events, all designations that suggest the attributed mechanisms for the clusters. Cluster 2 also contains the 4 stick-slips indicated as additional to the Pratt et al. (2014) catalogue. Clusters 3, 4, 5, 6, 7, 8, and 10 also contain small (i.e. < 5%) numbers of teleseismic events. Supporting information regarding the distribution of features per cluster and the features that most differentiate each cluster is provided (Fig. S7).

To further illuminate the underlying physical processes, we analyze the WIS clusters with respect to tidal heights computed at a downstream location (84°20'20.3994"S 166°0'0"W). This shows that Clusters 1 and 2 also vary with regard to occurrence at high and low tides, with Cluster 1 showing a slight tendency to high tide and Cluster 2 showing a more even split. This result is substantiated by previous studies on distinct modes of stick-slip (stick-slip variation in event length is dependent on initiation location, i.e. central or near the grounding line, and tide height; Pratt et al., 2014)). Cluster 5 events indicate the strongest tidal association with 75.1% of events occurring at high tides, while Cluster 9 events show a slight tendency to high tides.





Table 1: Characterization of clusters for $k$=10 by various measures: The percentage of events that are stick-slip events from the Pratt et al. (2014) catalogue (PRA14) and those additional to the catalogue (PRA14a); the percentage of events that are teleseismic events with amplitudes > 3.5 (log, a.u.; TI) and those with amplitudes ≤ 3.5 (log, a.u.; TII); the features that are most discriminant (most positive, + or most negative, -) and therefore most defining for a given cluster are listed in Table S5; the percent at which a cluster's events occur at high tide (the tidal heights used to gather the percentages are determined for a downstream location (84°20'20.3994"S 166°0'0"W) from the CATS tidal model (Padman et al., 2002; Howard, 2019)); the four seismometers that detect the majority of events per cluster, in order from most to least (Table S4); the days at which the most events per cluster are detected (Fig. S8 left column); the hours at which the most events per cluster are detected (Fig. S8 right column).

| Cluster | Events, % | Stick-slip events, % (PRA14, PRA14a) | Teleseismic events, % (TI, TII) | Most discriminant features in rank order (+/-) | Positive tidal, % | Spatial | Daily | Hourly |
|---|---|---|---|---|---|---|---|---|
| 1 | 7.49 | 84.2, 0.72 | 1.44, 0 | 20 (+), 14 (+), 68 (+), 24 (+), 34 (+), 16 (+) | 62.6 | BB04, BB14, BB03, BB06 | Intermittent | 7:00–18:00, 19:00–3:00 |
| 2 | 4.58 | 15.3, 3.53 | 8.24, 0 | 34 (+), 30 (+), 19 (+), 10 (+), 18 (+), 1 (+) | 50.6 | BB06, BB17, BB11, BB14 | Intermittent | 00:00, 4:00 |
| 3 | 9.16 | 1.18, 0 | 1.18, 1.76 | 1 (+), 21 (+), 71 (-), 10 (+), 15 (+), 37 (+) | 48.2 | BB17, BB10, BB08, BB13 | 7, 15, 18 | 23:00–5:00 |
| 4 | 6.41 | 1.68, 0 | 0, 0.84 | 4 (+), 19 (+), 20 (+), 22 (+), 10 (+), 21 (+) | 54.6 | BB15, BB06, BB01, BB07 | 2–5, 7 | 00:00–4:00, 14:00–16:00 |
| 5 | 12.3 | 0, 0 | 1.75, 0 | 27 (-), 29 (+), 17 (-), 30 (+), 21 (+), 24 (+) | 75.1 | BB16, BB15, BB12, BB01 | 33–34 | 1:00–12:00, 13:00–16:00 |
| 6 | 9.38 | 0, 0 | 0, 3.45 | 22 (-), 28 (+), 71 (-), 38 (+), 29 (+), 27 (+) | 63.8 | BB12, BB13, BB07, BB10 | 5–6, 33–35 | 9:00–19:00 |
| 7 | 10.6 | 0, 0 | 1.53, 2.04 | 36 (-), 13 (-), 37 (-), 35 (-), 68 (+), 3 (+) | 56.1 | BB13, BB01, BB07, | Intermittent | 00:00–5:00, 12:00–17:00 |

| | | | | | | BB10 | | |
|---|---|---|---|---|---|---|---|---|
| 8 | 5.71 | 0, 0 | 0, 3.77 | 13 (-), 4 (+), 36 (-), 35 (-), 14 (-), 34 (-) | 68.9 | BB12, BB13, BB07, BB10 | Intermittent | 1:00–5:00, 11:00–16:00 |
| 9 | 7.17 | 0, 0 | 10.5, 5.26 | 70 (-), 16 (+), 3 (+), 14 (+), 38 (-), 5 (-) | 64.7 | BB04, BB17, BB03, BB11 | 7, 11, 19, 30, 35 | 13:00–22:00 |
| 10 | 27.2 | 0.40, 0 | 0, 2.18 | 71 (-), 39 (-), 17 (-), 37 (-), 16 (-), 35 (-) | 59.0 | BB13, BB08, BB07, BB12 | Intermittent | 00:00–4:00, 14:00–16:00 |

We now explore the spatial and temporal distribution of events that are attributed to clusters (Fig. 6; spatial synthesis in Table S4; temporal synthesis in Fig. S8). From the spatial analysis, we observe that the seismometers detecting the most stick-slip events from Clusters 1 and 2 are located within a previously identified sticky spot (Fig. 6a, called the 'Central Sticky Spot'; Winberry et al., 2014; Barcheck et al., 2018) and a lesser proportion of events are detected near the grounding line, at seismometers downstream of a subglacial lake (Lake Engelhardt) located at 83.6°S 159°W (Fig. 6a Pratt et al., 2014). The temporal analysis highlights a unique type of event occurring on January 16th and January 17th, 2011, as all of Cluster 5 can be traced to these dates (Fig. 6b). Cluster 9 containing teleseismic events (about 16%) also contains events manually appraised as 'Other' that have similar high-energy signatures as the high amplitude teleseisms (Fig. S7). In the spatio-temporal attribution, we find that these events appear grouped in the time interval between 14:00 and 21:00 hours after 00:00 UTC. Preceding solar noon (22:00 to 23:00 UTC) this diurnal pattern could indicate a partial control due to changes in surface ice temperature (Fig. 6b).

## 4 Interpretation and discussion

We have demonstrated a step-by-step methodology for applying cluster analysis to a catalogue of seismic events recorded on a soft-bedded Antarctic ice stream. In this section we describe the limitations of this analysis with a view to informing further studies. We then outline the improved understanding of dynamic and hydrological processes at the WIS, and surrounding region, that this study has enabled together with the wider applicability of the methods and workflows.





## (a) Spatial attribution

## (b) Temporal attribution

**Figure 6.** The (a) spatial and (b) temporal attribution of clusters for $k$=10. The spatial attribution pie chart segment colors corresponds to the numerical cluster assignments provided in the key. The temporal attribution is provided as a daily occurrence for days after December 14, 2010 and an hourly occurrence for hours after 00:00 UTC, where solar noon falls in the time zone of 22:00 UTC. In conjunction, both attributions are used to better characterize the clusters for $k$=10. The results of the spatio-temporal analysis are synthesized in Table 1. A breakdown for the spatial attribution and temporal attribution is provided in the Supplementary Materials (Table S4 and Fig. S8, respectively). Colors were chosen for clarity between separate classes (Glynn and Naylor, 2021, Ordnance Survey). The spatial and temporal attribution of clusters for $k$=10 showing high confidence events only is included in Fig. S9.





## 4.1 Feature selection

The elimination of highly correlated and/or inconsequential features prior to clustering is an important step to reduce input bias. In a supervised learning application, such as Provost et al. (2017), feature selection can be guided by the quantified error in clustering output based on a labelled, expected result (Breiman, 2001). In the current study, we carried out an evaluation to determine which features are most discriminant using a correlation analysis (Sect. 3.1.1). Further, it can be informative to examine how the clusters change if shown a similar, alternative feature set.

We compare the stability of the clusters that result with $k$=10 by clustering events using: a) our standard feature set, b) all of the features from Table S1 (no selection and elimination) and c) a halved feature set (Fig. 7). This result shows that the no elimination option results in about half of the clusters becoming less-defined in comparison to the standard feature set. The halved feature set comprises 17 of the less correlated feature pairs from Fig. 3, i.e. a maximum number of features are eliminated, in comparison to the standard set where a minimum number (i.e. only the most correlated) are eliminated. The

percent composition of events in the halved set is comparable to the standard, but again, some redistribution of events occurs, significantly impacting about half the clusters. The distribution in clusters of the positively identified glacier processes remains stable in the halved feature set, so it appears to be more useful to eliminate too many features rather than too few.

We also evaluate the stability shown in Fig. 7 with the high confidence events only (Fig. S10). This result is used as an additional way to discuss the robustness of each cluster. We find that Clusters 1, 2, 3, 5, 6, and 9 all appear well-defined (i.e.

more robust) across the other feature set scenarios. The other clusters (4, 7, 8, and 10) indicate higher instabilities (i.e. less robust) in how the contained events are partitioned. Such a result provides insight into other dimensions of similarity across the noise-type events that conflict with the current feature domain boundaries.

## 4.2 Cluster number and assignment of process

The choice of the number of clusters is understood as a significant challenge in the application of k-means++ because it is

difficult to fix a number that captures all the levels of variability in a dataset (Hardy, 1996). While many studies have explored a plethora of approaches, ultimately, the decision requires knowledge of the user domain and is therefore problem-dependent (Jain and Dubes, 1988). We recommend that future studies make use of the methodology, assessing how clusters split or remain stable (Sect. 3.3) to best understand the groups of events represented in each cluster. Such a framework provides a way to ascertain that clusters, or parts thereof, capture distinguishable, physical mechanisms of ice stream dynamic and hydrological

processes.

The semi-automated approach combines the power of unsupervised learning with information from prior or independent manual analysis, to inform the interpretation of clusters. For example, the tidal influence on stick-slip events is a previously identified relationship between seismicity and temporal cycles (Winberry et al., 2014), and this study extends that understanding. We infer that these events exhibit two different characteristic lengths and occur at the Central Sticky Spot and the

grounding line (Fig. 6). We are thus enabled to investigate how the spatio-temporal association of these events can relate to tidal influences.



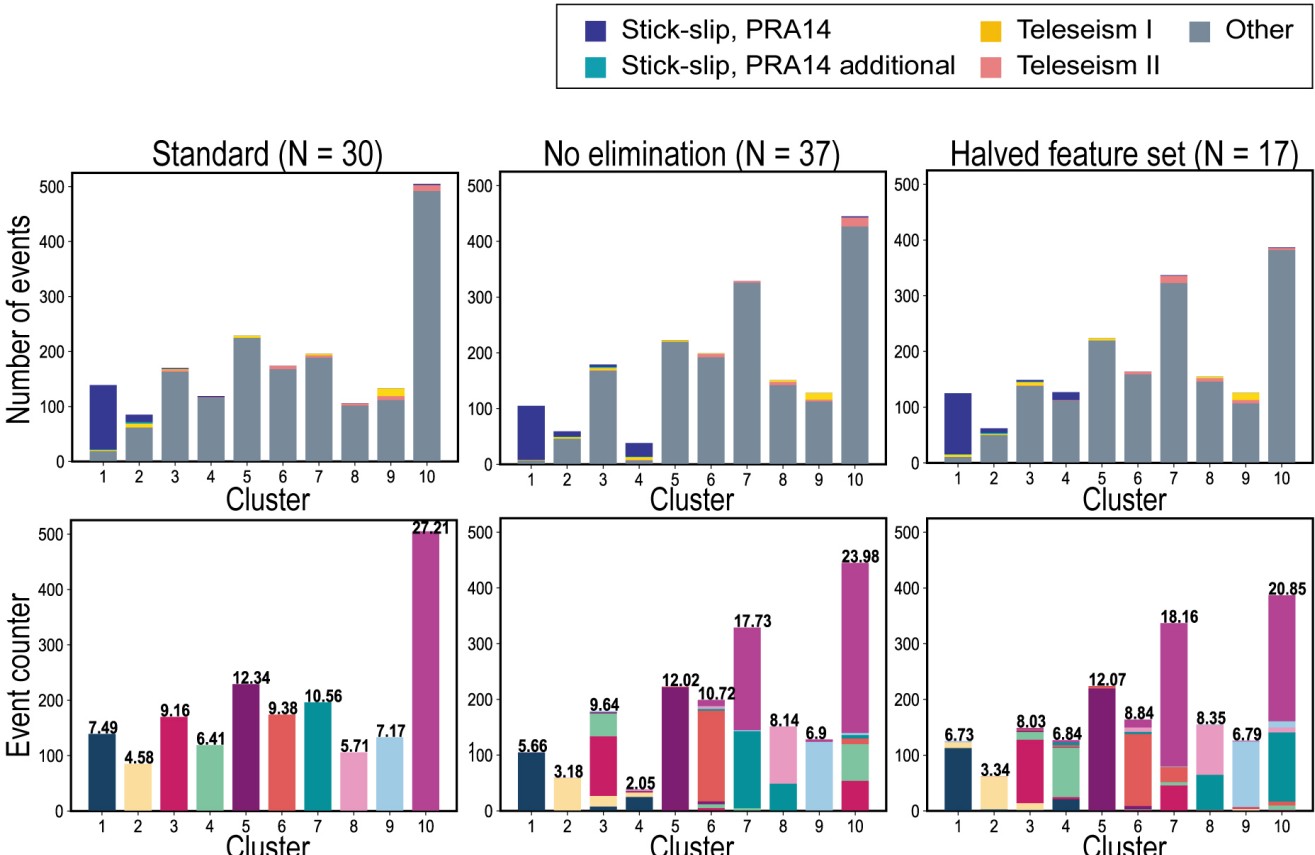

**Figure 7.** Stability of cluster results compared with the standard procedure. The left column shows the standard set of features after removal of selected members of the feature pairs with r⩾0.95. The middle column shows the results of clustering with no features eliminated and the right column shows the results with a halved feature set. The top row provides the percentage composition of each cluster 1–10 as determined from the manual appraisal. The bottom row provides the number (and percentage) of events that are grouped in each cluster. The order and labelling of the clusters in the middle and right columns is manually decided by a comparative method that matches clusters approximately one-to-one for each application, when possible. The colors within each cluster in the bottom row signify where the contents of the clusters resulting from applying the algorithm to the standard feature are placed in the two comparison applications of the algorithm. The stability of cluster results compared with the standard procedure showing high confidence events only is included in Fig. S10.

The heuristic technique that we have used in this study has enabled the pre-existing understanding of glacier processes active on the WIS to be extended, and has also newly enabled the discovery of other active processes in the WIS dataset that match to mechanisms reported elsewhere. These results confirm the value of using an unsupervised learning approach.

**4.3 Identified glacier processes and other signals**

In our demonstration study, 136 stick-slip events have been identified previously in the WIS dataset (Pratt et al., 2014) and 4 are found in addition by the event detection and semi-supervised learning workflow. High energy stick-slip events that mainly initiate from the Central Sticky Spot during high tide, on a diurnal cycle, are contained in Cluster 1 (Fig. 5). A lesser proportion





of these events initiate closer to the grounding line and occur during low tide. The Cluster 1 events correspond to both first
and second ruptures of stick-slip, a result consistent with the initiation locations expected for first ruptures (Central Sticky
Spot or grounding line depending on high and low tides, respectively) and second ruptures (near grounding line). Stick-slip
events contained in Cluster 2 are of higher energies and mostly span both the first and second ruptures (i.e. are longer in total
duration). These events are associated with both high and low tides and are detected evenly across the Central Sticky Spot and
grounding line, likely because of the even influence of tidal control. The Cluster 2 type of stick-slip events are grouped in the
same cluster as comparatively long and energetic teleseismic events (of type Teleseism I).

Further newly identified events occur in the hours preceding solar noon with an irregular periodicity of 4–11 days. These high
energy events (detected as low-strength) propagate horizontally from far-field, and group with events (16%) manually identified
as teleseismic events in Cluster 9 (Fig. 5). From the feature-based analysis, we suggest that these signals could be generated
external to the WIS, potentially from the Ross Ice Shelf. Although the signals that we observe have longer frequencies, the
mechanism could be related to contraction due to cooling, or to cracking triggered by stress from local melt (Hudson et al.,
2020). The Cluster 9 non-seismic events therefore suggest mixed diurnal influences (MacAyeal et al., 2019).

We detected a swarm of low energy, high frequency events that are unique to two days of the deployment: January 16th and
17th, 2011. This event swarm is contained in Cluster 5 (Fig. 5), and shows a spatial pattern consistent with the events occurring
near the grounding line (Fig. 6), and a temporal pattern linked to high tide (75.1% occurring at a positive tidal anomaly; Table
1). The lower energies of these events and the relative high frequencies (Fig. S7; Table S5) could indicate that these events
originated external to the WIS and so the energy was relatively attenuated by time of arrival (Pérez-Campos et al., 2003). The
association with the grounding line and tidal cycles could indicate that these events originated on the Ross Ice Shelf, where
tidally induced seismicity has recently been recorded near the ice shelf grounding line (Cole, 2020) and high frequency ambient
resonance during a several day period can be caused by a surface melt event (Chaput et al., 2018; Jenkins et al., 2021). Process
swarms of this nature require a detection mechanism with a high temporal resolution, and show the advantage of using seismic
methods as part of the tool box for understanding glacier processes.

The minority of detected events are related to the dynamic glacier processes; the rest are related to the noise field external to
the glacier (Fig. 5). The diversity in the noise events is of significant interest as this informs the understanding of cryoseismic
and ambient seismic processes in the wider Ross Ice Shelf region. Understanding the diversity of signals is also of utility
in understanding the output of the multi-STA/LTA algorithm (used to generate the underlying event catalogue) in terms of
identifying events as local to the ice stream, or external to the ice stream. Events characterized by long durations, high energies,
and tremor-like behavior that occur mainly on several days in the beginning of the deployment period in December, 2010, make
up Cluster 3. Cluster 4 events have similar temporal structure and high energies but slower emergence, preceding the tremor-
like behavior and overall shorter durations. Short, energetic signals that appear mostly between January 16–18, concurrent with
the Cluster 5 icequake swarm, are represented in Cluster 6. Noise-type events that have similar spatio-temporal occurrences
are grouped into Clusters 7 and 8, with the lower energy events contained in Cluster 8. Cluster 10 contains events with low
energies across all frequency bands, spatial associations with the grounding line, and diurnal patterns intermittently through
the deployment. Collectively, the noise-type ambient signals in these clusters, likely related to processes from the nearby Ross



Ice Shelf and Ross Sea, are heterogeneous, but cluster in a range of frequencies ($10^{-1}$–1 Hz, Fig. 4). This result is useful for

guiding future monitoring applications.

As a final comparative summary, the domain of each cluster in the duration-frequency space is shown in Fig. 8 as a snapshot of the Whillans Ice Stream seismicity, in terms of labelled event types (e.g. stick-slip and teleseism) and the other process and noise related events made possible by unsupervised learning. An overview of each cluster, most likely source mechanism, and other comments are provided in the Supplementary Materials (Table S6). The manual appraisal has provided a foundation for

identifying further event types and their domain boundaries in duration-frequency space. Figure S11, showing high confidence events only of Fig. 8), offers further insight into the groupings of duration-frequency domains. For example, high confidence Cluster 9 events not labelled as teleseisms primarily occupy an intermediary duration-frequency domain, of 20 –120 seconds and 0.5–0.1 Hz.





**Figure 8.** Synthesis of the Whillans Ice Stream duration and frequency relationships by manually identified event types and unsupervised learning clusters. The manually identified events are designated by filled circles. The clusters are shown in the two plots as (a) Clusters informed by manual appraisal and other reconnaisance, including Cluster 1 (stick-slip), Cluster 2 (stick-slip), Cluster 5 (potentially melt pulse swarm), and Cluster 9 (teleseismic and potentially external fracture-related), and (b) Clusters likely pertaining to noise-type events, including Clusters 3, 4, 6, 7, and 8. Some event types can be concealed by other symbols; the events of type 'Other' and other manually identified events are arranged in the background, such that the unsupervised learning result is in the foreground. The synthesis of the Whillans Ice Stream duration and frequency relationships in Fig. S11 shows only high confidence events.



## 4.4    Applicability of approaches

The workflow presented enables a cryoseismic event catalogue, previously built using a straightforward, systematic detection algorithm (Latto et al., 2023), to be analyzed using a similarly transparent approach to a data-driven investigation, i.e., using k-means++ and analysis of clusters. While the workflow has been demonstrated for the Whillans Ice Stream, this is a widely applicable approach, which could be used as a benchmark procedure for glacier investigations and monitoring, and many other environmental seismology applications. The procedure is well-suited to cases where the recorded seismicity consists of varied

event types in a similarly varied ambient wavefield. The workflow requires a manual appraisal of events, resulting in plots (Fig. 4) that show an overview of the event populations that are present and reveal useful domains of the feature space. This information is used, as has been demonstrated (Fig. 5; Table 1), in tandem with the unsupervised clustering to understand how the events due to glacier processes are held within the automatically identified clusters.

As one application of the method, comparisons could be made for one glacier from year to year, making use of the very

fine temporal resolution and ability to detect hidden processes that is the advantage of cryoseismic approaches. After the semi-automated reconnaissance of the first year (manual and unsupervised approaches being used together), this work can be used in subsequent years to streamline unsupervised learning with a greater proportion of automation. Thus it would be possible to undertake a sequence of studies in time, year after year, to detect changes in the spatial and temporal distribution of known processes and to identify new, potentially hidden processes.

The standardization of the method also enables for robust comparisons between locations, with bivariate feature plots (Fig. 4) forming a basis for such investigations. Comparative analyses could inform the similarities and differences between event populations in each location and could be further developed using the framework for semi-automated analysis used herein. The clustering of events recorded on the WIS, including event mechanisms that are local to the glacier and also those that are part of the surrounding seismic ambient noise wavefield, could be used as a guide to what might be expected at other localities

and/or from other deployments.

## 5    Conclusions

Motivated by the potential for data-driven approaches to characterize the diverse signals recorded from glacier environments on the margins of ice sheets, including both local events and ambient noise signals, we have presented a workflow that makes use of semi-automated learning. As a demonstration example, we have shown how glacier processes may be investigated using

seismic events recorded on the Whillans Ice Stream. We use a near-comprehensive event catalogue of the 2010–2011 austral summer (companion paper, Latto et al., 2023), from which we form a dataset that represents each event by a selected set of 30 seismic time series features (attributes). Application of an unsupervised learning algorithm, k-means++, to the feature dataset produced ten best-separated clusters (groups with similar characteristics) of seismic events. These were analyzed with the benefit of a manual appraisal of the catalogue. Although the clusters are only moderately separated, they nevertheless enable

valuable reconnaissance analysis.



We identified the following glacier processes: high energy stick-slip events that mainly initiate from the Central Sticky Spot, longer and higher energy stick-slip events initiating near both the Central Sticky Spot and grounding line, high energy events that propagate horizontally from far-field generated as part of the external wavefield with a diurnal pattern. The latter group of signals could be caused by processes on the Ross Ice Shelf such as contraction due to cooling, or to cracking triggered by stress

from local melt. We found a swarm of high frequency events that are unique to two days of the deployment: January 16th and 17th, 2011, and suggest a relationship to a Ross Ice Shelf melt pulse. The moderate tidal control on the overall event catalogue is seen in the stick-slip events (as previously identified by Pratt et al., 2014) and also the swarm pattern, and is also evident to some extent throughout the large number of noise-like events in the catalogue (Latto et al., 2023).

The majority of detected cryoseismic processes are likely related to the complex noise field external to the glacier. By

means of clustering, a reconnaissance analysis is enabled, showing signals with a diversity of durations, energies, and behavior (e.g. tremor-like). The variability we find in the ambient noise illustrates an additional challenge for cryoseismic analysis. It is necessary to separate the multiple different local glacier processes listed above from the diverse signals and changing amplitude levels of the seismic ambient wavefield. For the Whillans Ice Stream, noise signals cluster within a range of frequencies separated from the domain occupied by some of the process-related events, between $10^{-1}$–1 Hz. Clustering approaches will

enable automated methods to be used with a much smaller component of human analyst review in future monitoring of the Whillans Ice Stream using seismic methods.

The demonstration workflow enables a transparent approach to data-driven investigation of glacier processes with seismic signals. The potential for applying unsupervised learning algorithms to cryoseismic signals is vastly improved by a methodical assessment of cluster attribution. That is, future investigations can take better advantage of unsupervised learning methods,

following such an initial semi-automated appraisal that includes a manual component, so that the semi-automated methodology can continue to identify new event types on a glacier from year-to-year. An extended investigation into the diversity of local events and ambient noise from a given glacier seismic deployment would enable glacier monitoring, using seismology, for that key location. Future investigations could take advantage of the new methods and workflows that we have presented, which should prove widely applicable to identifying changes in glacier environments, and comparing locations, as they respond to

the changing global climate.

*Code and data availability.*    The trace and reference seismic event catalogues (Sect. 2) and the trace and reference feature datasets (Sect. 3.1) are made available in the Electronic Supplement. Both are produced by extensions to ObsPy software made available at: https://github.com/rossjturner/seismic_attributes by Turner et al. (2021). The results of the k-means analysis (Sect. 3.2) are also provided in the Electronic Supplement by a merged reference catalogue with selected features and cluster labels. Related data for analysis is publicly

available: The Whillans Ice Stream seismic dataset is accessible from The IRIS Data Management Center (IRISDMC) (Winberry et al., 2010). The Tide Model Driver (TMD) toolbox allowed for appropriate use of the Circum-Antarctic Tidal Simulation (CATS) (Padman et al., 2002; Howard, 2019) to compute tidal heights at the Whillans Ice Stream.



*Author contributions.* RBL developed software, carried out data analysis, and wrote the text; RJT developed software and provided guidance; AMR gave overall project direction and provided guidance; SC, BK, and JPW advised on the dataset and provided background context. All
authors contributed to the refinement of the text.

*Competing interests.* No competing interests are present.

*Acknowledgements.* This research was funded under Australian Research Council Discovery Project DP210100834, with additional support from DP190100418, ARC's Special Research Initiative The Australian Centre for Excellence in Antarctic Science, SR200100008; and the Australian Antarctic Program Partnership. In addition to the newly available code as above, we used software from the ObsPy Python project
(Beyreuther et al., 2010; Megies et al., 2011; Krischer et al., 2015), Fig. 1 was generated by *agrid* Python module (Stål and Reading, 2020) with assistance from Tobias Stål. We also wish to thank colleagues Tobias Stål, Hannes Hollmann, Ian Kelly, and other UTAS/IMAS group members and collaborators for contribution to discussions. We thank the two anonymous examiners of the Master of Science Thesis (for Rebecca Latto, Author 1), in which this research has been included as a core chapter, for their careful review of the material and thoughtful suggestions for improvement.



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
