# Peer review of "Towards the systematic reconnaissance of seismic signals from glaciers and ice sheets - Part B: Unsupervised learning for source process characterisation"

_EGUsphere, 2023_

## Author Comment (AC1)

Title: Towards the systematic reconnaissance of seismic signals from glaciers and ice sheets - Part B: Unsupervised learning for source process characterisation
Author(s): Rebecca B. Latto et al.
MS No.: egusphere-2023-1341
MS type: Research article

Please access the discussion at:
https://egusphere.copernicus.org/preprints/2023/egusphere-2023-1341/#discussion
* * *
Below, the R1 comments are copied in grey. Author Comments continue in blue.
* * *
**General Comments**

The manuscript presents a thoughtful and accessible methodology for performing clustering analysis for glacial seismology. Given increased interest in continuous seismic monitoring of the cryosphere, the study is timely and instructive and would be of value to members of both the seismological and cryosphere communities. Furthermore, the manuscript is well written and strikes an appropriate balance between introducing basic theory and demonstrating careful application and thorough analysis.
 Many thanks for this positive appraisal, and an expert review that probes the analysis (including the supplement) very thoroughly.  The contribution will benefit significantly from clarifications and additions suggested, and we are happy to follow all suggestions made.

One of the chief concerns I have is the utility of k-means on a parameter space with 30 dimensions. Though the authors took care to appropriately select features for clustering, the procedure was not conceived with dimensionality reduction in mind, nor is dimensionality reduction mentioned once in the literature review, although several papers were cited that make use of it (e.g., autoencoders, PCA). Given the dubious utility of distance-based clustering metrics in high dimensions (Aggarwal et al. 2001; Aggarwal & Reddy 2014), I would at least expect acknowledgement of this limitation. I hesitate to require re-performance of what is already a substantial analysis presented in this manuscript, but a good candidate for future work would be repeating the analysis but with reduced dimensionality, e.g., using the top 5 or 10 components from PCA.
 We agree that the reviewer and suggested references articulate important considerations, and we are happy to include a new paragraph in the interpretation and discussion section that addresses these points.

My other main critique is the use of the k-means(++) algorithm. K-means has several important limitations, namely that it performs poorly with overlapping clusters and that it assumes equal variance in all dimensions. It is sensitive to outliers, which is exacerbated by the high dimensionality of the feature space. I would instead suggest using an expectation-maximization (EM) approach to refine the cluster definitions, as provided in Gaussian mixture model (GMM) clustering. GMM handles multivariate distributions and overlapping clusters better than k-means, and is trivial to implement with scikit-learn. However, GMM is still subject to the curse of dimensionality.
 We made the choice to use k-means(++) based on its transparency for a reconnaissance workflow.  We

are happy to note this, and also include the points raised in the interpretation and discussion section, and are grateful for the suggestion to point readers to an alternative unsupervised algorithm (GMM), and its own strengths and limitations.

In summary, the authors should at a minimum revise the manuscript to acknowledge the curse of dimensionality and the limitations of k-means clustering. Reworking the entire analysis to reduce dimensionality and implement GMM clustering, while desirable, is likely beyond the scope of this work. Agreed, noting that the transparency of the algorithm that we use is a reasonable fit to the scope of the study.

**Specific Comments**

[Line 62] I would be careful with your use of "high-dimensional" here. As you correctly identify, *k*-means clustering relies on Euclidean distances; however, Euclidean distance becomes a less useful metric for clustering since, with increasing dimensionality, data become sparse and distance less meaningful (the curse of dimensionality). It thus becomes increasingly difficult to distinguish clusters, since all the distances between points appear the same. An informative exploration of this phenomenon is given by Aggarwal et al. (2001, http://link.springer.com/10.1007/3-540-44503-X_27). We'll re-write this paragraph, removing the phrase from the first sentence, and adding the suggested caution (which we agree with) later in the paragraph. We'll also add the reason for our choice of algorithm and note one alternative (GMM) as suggested above.

[Line 96] I would suggest citing the paper (Jenkins et al. 2021), not the AGU abstract. Thank-you for picking this up. This reference will be updated.

[Line 102] You cite another AGU abstract by Sawi et al. - if they have a related paper, you should cite it instead. This reference will be updated to the full paper also.

[Lines 170-176] I appreciate the thorough explanation, but feel it can be said more concisely. We're happy to edit accordingly.

[Line 175] "...the features demonstrate comparable distributions." Do they? Perhaps you mean to say they are distributed over comparable scales? Because the distributions themselves are quite unique: some are normal, some bimodal, etc. We're happy to clarify as suggested.

[Line 178-179] Let me first say that I am pleased to see this careful treatment of your input features, acknowledging that not all features are useful for the clustering analysis. However, I think you should explain your reasoning further or at least provide a citation to a useful reference, for the benefit of your readers. What type of bias are you referring to? Is it distinct from the bias you strive to eliminate by standardizing your features? Why is the inclusion of too many features bad? We will clarify these lines, in the context of the paragraph before them where we define bias in a slightly different use case. We will remove the word *bias* in Line 178 as it is misleading.

[Sec. 3.2] Doesn't k-means as implemented by scikit-learn use random restarts, as well? If so, you should include this in your description and discuss how the centroids are determined accordingly. We will add a sentence on this point. The clusters may organise in a different order, but we can identify matching clusters in this study through some (manually) identified event groups.

[Lines 207-208] The silhouette score is tricky to describe, and I'm afraid your explanation leaves me confused. Particularly confusing is the phrase, "each cluster set of means." Please clarify the explanation.
 We're happy to provide a more condensed explanation.

[Sec. 3.1, Sec. 3.2, Fig. 3] You index clusters by $r$, but also use "r" as the correlation coefficient. I would suggest de-conflicting your notation, e.g., indexing clusters as $k=1,...,K$ and reserving $r$ for the correlation coefficient. Furthermore, I would suggest italicizing the correlation coefficient so it is abundantly clear you are referring to a parameter. In the caption of Fig. 3, I initially interpreted "(r)" as "right-hand."
We will keep the k as used because of its prominence in k-means literature, but we agree to change the r notation as it conflicts with correlation coefficient. We will plan on substituting the integer value of r to g in the k-mean steps detailed in Sect. 3.2. Other options conflict with variables widely used in seismology, but we defer to the editor if there is a better suggestion.

[Line 262 & Suppl. Line 46] Why is $k=14$ a "realistic" maximum?
 This is a reasonable number of distinct event types, for the complex seismic wavefield of the WIS.  For a valley glacier (for example), without the adjacent ice shelf, we anticipate that a smaller k value would be more appropriate.

[Sec 3.3.2, Sec. S3.3, Fig. 5, Fig. S6] I applaud your effort to thoroughly examine the "evolution" of clusters. However, as currently written, the first paragraph of Sec 3.3.2 leaves me with a sense of confusion at best, or at worst that the analysis may be flawed. How are you able to track the progeny of $k$ clusters from from the previous $k$-1 clusters? You mention that k-means has built-in pseudo-randomness (and random restarts), but between Section S3.1 and this section, I fail to understand how (and why) you overcame this. Are you setting the seeds manually, and preventing restarts? Though you refer to Section S3.3 in a previous section (Line 209), I would refer to it again here, as there are crucial details in it that you should relate to this part of the text.
 We agree this point needs clarifying and are happy to do so.  In our study, we can use event groups to (manually) identify matching clusters (typically they are not in the same order, but have recognisable content) but other studies should consider controlling the re-starts as you mention.

The last thing I'll say about this section & related supplement/figures is that I had to re-read them several times to become convinced that you are not proposing that the composition of clusters at $k$+1 is dependent on the composition at $k$. The discussion in paragraph 2 (Line 261) ultimately settled the question, but I think if you clean up the first paragraph, you can alleviate the confusion. With the question settled, Fig. 5 is a nice analysis of how cluster composition changes.
 We are happy to clarify as suggested.

[Suppl. Line 64] This is not *a priori*; it is *a posteriori* since you must run experiments to determine the optimal number of clusters.
 We'll delete that word.

[Suppl. Line 66]  "that is, ..." This is self-evident and redundant.
 We'll delete that half-sentence.

[Suppl. Lines 65-71, Fig. S3b] What justifies ignoring the outliers?
We will clarify the sentence at line 66 as it is confusingly worded. We will also remove the last two sentences as the quantitative information is displayed in Fig. S3.

[Fig. 7] The top row is an interesting analysis, but I'm not convinced of the value of the bottom row. To me, it seems you are comparing apples to oranges. Because the datasets are essentially different due to the inclusion/exclusion of additional features, there are too many factors that affect the cluster assignment of data, including pseudo-random seeding, restarts, and, even if those are controlled, the vastly different dimensionality of the parameter space. All this is to say, clusters in one dimensionality do not look like clusters in another dimensionality, as indicated by the top row of subplots.

We agree in general (noting previous clarifications above), and are happy to qualify the text that refers to the bottom row. We do think that it is useful to include to illustrate the variability of results given choices made in the workflow.

**Technical Corrections**

[Line 101] A space is missing between "fracture" and the citation.
Space to be added.

[Line 182] Just use "variance."
Wording to be changed.

[Fig. 2] This is a rather trivial comment, but your 4th column is missing x-axis tick labels. My preference (certainly not a prescription) for this type of figure is to have the same y-scale for all plots, and print the axis labels on just one subplot.
Tick labels to be added. We prefer to retain the annotations on all columns, as the tick labels are very simple and give a visual confirmation that a consistence scale is used on the x axis.

[Suppl. Line 58] Change to "where $n$=100 is the number of bins...", etc.
Wording to be changed.

---

## Author Comment (AC2)

Title: Towards the systematic reconnaissance of seismic signals from glaciers and ice sheets - Part B: Unsupervised learning for source process characterisation
Author(s): Rebecca B. Latto et al.
MS No.: egusphere-2023-1341
MS type: Research article

Please access the discussion at:
https://egusphere.copernicus.org/preprints/2023/egusphere-2023-1341/#discussion

Below, the R1 comments are copied in grey. Author Comments continue in blue.

I have read with great interest the scientific article titled "Towards the systematic reconnaissance of seismic signals from glaciers and ice sheets - Part B: Unsupervised learning for source process characterisation" submitted by Latto et al., for publication in *The Cryosphere*. In this article, the authors propose to systematically explore seismic data acquired by a network of stations deployed on the Ross Ice Shelf during the austral summer of 2010-2011. The data processing pipeline, based on a priori detection (detailed in a companion paper), relies on the clustering of seismic events through the deployment of the K-means clustering method on a feature space computed from curated seismic signals features. The authors discuss the influence of feature selection and the number of clusters (one of the hyperparameters of the K-means method) on their results. They demonstrate that this approach is at least capable of revealing relatively pure clusters containing microseisms generated by stick-slip phenomena associated with the dynamics of the ice shelf. This clustering also allows for the identification of new microseismic events associated with tidal forcings.

The paper proposed by Latto et al. is remarkable for several reasons. Firstly, it is very well written and easy to follow. The literature review is particularly relevant while remaining concise. All critical information is contained within the paper, but the authors also provide a significant amount of supplementary results that address questions the reader may have. The division between the main content and supplementary content is particularly relevant. The results are convincing, and the discussion on clusters, methodology, and especially the choices of hyperparameters and features is comprehensive and very interesting. The figures are of good quality, although a few minor improvements could be made (see my comments below). Overall, this is an excellent contribution that will undoubtedly have a significant impact on communities interested in cryo-seismicity and other applications in environmental seismology. Therefore, I strongly recommend the publication of this article. However, I do have some minor comments that I will detail below.
Many thanks for this warm appraisal.  We're happy to address the general and minor comments as given.

The choice of the K-means clustering method appears appropriate for the present study, and the following suggestions are by no means an invitation to completely revise this paper. I believe it is already comprehensive and insightful enough for publication as it is. However, I would like to draw the authors' attention to the fact that this clustering method may not be the most relevant for working with the features proposed by Provost et al. (2017). These features often have values distribution overlapping  between each class of events, but K-means is not able to consider the "fuzzy" boundaries between different cluster. Methods

such as Gaussian Mixture Models seem to be more suitable, and I suggest that the authors at least explore this family of methods in future work.

We will add a note that we chose k-means++ due to its transparency of usage.  We agree that GMM is a sensible suggestion and will incorporate this suggestion into our response to Part B R1.

The choice of the clustering method used, however, is secondary compared to the more important question of feature selection. In this article, the authors propose to reduce the number of features by calculating pairwise correlation coefficients between features. This step is absolutely necessary for K-means to work well (as demonstrated in this study). However, this requirement limits the robustness and versatility of the approach proposed. The chosen set of features may work well for this dataset, but the correlations may be different for another dataset. Furthermore, even correlated features can carry complementary and, more importantly, relevant information for discriminating certain events (in supervised approaches like Random Forest or Gradient Boosting, removing correlated features usually reduces precision scores). An alternative approach to reducing the number of features would be to reduce the dimensionality of the feature space using dimensionality reduction methods, which would eliminate the cross-correlation selection step and retain some of the information carried by each feature. I encourage the authors to consider the possibility of using methods like PCA (although it rarely works on seismic data), or even better, t-SNE (Van der Maaten and Hinton, 2008) or UMAP (McInnes, Healy, & Melville, 2018) for future work.

Many thanks for this suggestion.  We'll include the information given in the discussion section.

The question of feature normalization is also critical. The authors chose to normalize by the standard deviation. This normalization helps to homogenize the overall distribution in the feature values space, but it sacrifices some level of information about each feature. The absolute value of the feature is probably as important as the position of the feature value in the overall distribution. Figure 4 presented in this paper is a clear example of this. What would your clustering result look like if you applied K-means (or another method) with non-normalized values in a two-dimensional space (Characteristic frequency - Duration) or three-dimensional space (Characteristic frequency - Duration - Peak Amplitude)?  Have the authors tested their approach without normalization? Finally, why was standard deviation chosen for normalization? If the goal is to preserve the properties of the distributions of each feature, there are other normalization approaches that may be more relevant (Yeo-Johnson transform, Quantile Transform, Unit Vector Scaling, Sigmoid scaling?). I suggest testing these in future work.

We are happy to add a short note of clarification on this point.

We agree that k-means without normalisation can homogenise feature distribution and is certainly a factor to consider in terms of the high-dimensionality curse (Aggarwal et al.,. 2001). We, however, scaled the data such that each feature was not necessarily transformed to a normal distribution (Fig. 2), thereby providing critical diversity to the inputs to our k-means++ unsupervised clustering. We agree that in some cases, clustering algorithms that are more predictive (e.g. that in Provost et al., 2017, a key reference in the m/s) can benefit from a different treatment of features.

Aggarwal, C.C., Hinneburg, A., Keim, D.A. (2001). On the Surprising Behavior of Distance Metrics in High Dimensional Space. In: Van den Bussche, J., Vianu, V. (eds) Database Theory — ICDT 2001. ICDT 2001. Lecture Notes in Computer Science, vol 1973. Springer, Berlin, Heidelberg. https://doi.org/10.1007/3-540-44503-X_27

In the first stages of the m/s development, we did try k-means without normalization. But, because of the large magnitude differences found that results were skewed such that limited information about glacier processes were discernible (e.g. in one case clusters were only bounded by energy levels, which were features of the highest magnitudes between 10^4 to 10^10). We then chose standard deviation as a transparent approach to scaling features, but are happy to note an example of the other suggestions in the discussion. Particularly, the quantile transform would be an interesting method to consider.

Minor Comments :

L.102 : A missing space "fracture(Hammer) [...]"  Will be corrected.

L.163-164 : This needs some clarification : you computed the median, the mean or something else of the values of the features of the seismic signals recorded at each station for a given event ?
We agree that this line requires clarification and citation. Feature calculations are more explicitly expounded upon in the companion paper (Part A: Latto et al., 2023a) and in our reference work (Turner et al., 2021). First, for a given event, we compute and sort trace catalogue seismometer records in order from most energetic (i.e. largest maximum peak amplitude of a record) to least. For that event, feature values are chosen as the median — or second highest — from the top 3 most energetic records. Please refer to the supplementary Table S1 for feature-specific information where this general formula can vary per feature.

Latto, R. B., Turner, R. J., Reading, A. M., and Winberry, J. P.: Towards the systematic reconnaissance of seismic signals from glaciers and ice sheets – Part A: Event detection for cryoseismology, EGUsphere [preprint], https://doi.org/10.5194/egusphere-2023-1340, 2023.

 Turner, R.J., Latto, R.B. and Reading, A.M., 2021. An ObsPy Library for Event Detection and Seismic Attribute Calculation: Preparing Waveforms for Automated Analysis. *Journal of Open Research Software*, 9(1), p.29.DOI: https://doi.org/10.5334/jors.365

Have you also considered adding in your feature arrays the standard deviation of the values of the features of the seismic signals recorded at each station ? This can provide some valuable information on the location of the source and on other geometrical properties of the event (e.g. near of far field origin, dip, etc.).
We are happy to add to the discussion regarding use of the feature information as suggested.

L.175 : If for each feature you have the same distribution then those features become useless for any identification methods. Please rephrase and clarify.
We agree that the m/s insinuation that the features have the same distributions is misleading. As noted above, the features are scaled (not normalized– as in not transformed into normal distributions necessarily). The features are transformed by standard deviation to be between -4 and 4, but otherwise have diverse distribution shapes (Fig. 2), lending to interesting clustering results.

L.207 : There are other estimators used to try to determine the relevance of a clustering result besides the Silhouette test (e.g. Davies-Bouldin score, Elbow score, Calinski-Harabasz index, Dunn index). It would have been interesting to see how these factors evolve in relation to the number of clusters chosen and in comparison to the ideal number decided by you.
We agree that we can note in the paper that there were a variety of estimators to choose from. In our preliminary work for this m/s, we considered each of these other estimators along with the Silhouette test for their individual definitions of *similarity* to help guide the choice of k. However, ultimately, we decided to construct our own test for measuring similarity because these typical scoring methods were difficult to interpret in our high-dimensional feature space where *separation* vs *similarity* among 30+ features is convoluted to quantify. We chose to reference and show the results of the Silhouette test as a widely-used assessment that could provide an established point of comparison for our new test but agree that the evolution of cluster numbers as determined by other estimators could be a future avenue of investigation.

L.231-232 : You find 136 stick-slip events identified as such by Pratt et al. (2014), 4 new ones, but how many did you miss from the Pratt et al. (2014) catalog ?
We are happy to clarify as this is not stated in this m/s (but covered in Part A). We did not miss any of the stick-slips in the Pratt et al., 2014 catalogue.

L.241-243 : Indeed, but don't you lose this information by your normalization of the features (see general comment)? The duration/peak amplitude correlation also seems quite strong (as is often observed for microseismic sources, which are sometimes assigned a "duration magnitude"). Is one of these two features therefore excluded from your analysis?

We are happy to add a note in the earlier section that subsequent comparisons refer to normalised values where appropriate. And also a sentence or two in the discussion about possible information loss at various stages of the workflow.  We agree that this would be helpful to others using the workflow.

Figure 4 and Figure 7 miss subplot labels "a", "b", "c", etc.

Thank you for this careful read of the figures. In Fig. 4, our intention was to leave off subplot labels and they are not referred to in the text. However, we can follow editor guidance on this point. We intended the same for Fig. 7, though see now that subplot labels are referenced on lines 315–316. Based on editor guidance, we can either clean up the text such that (a)--(c) are not referenced as such, or we can add subplot labels to the figure and caption.

Figure 6 (b) : The X-axis label "Days after December 14, 2010" is not convenient for interpretation I think. I suggest giving the real dates.

Our intention with the x-axis label here was to be able to assess periodicity over the two months, and use the # of days as a value to refer back to in the Discussion. However, we also agree with this suggestion, and can add annotation to certain dates (e.g. Jan 1, Jan 31), to provide more information such that the reader does not have to add from Dec 14 to investigate a date of interest noted in our interpretation (e.g. Jan 15).

L.317 : Define "less-defined". Maybe quantify this (percentage of events from a given class?)

As this is not central to the presented work, we prefer to retain the qualitative comparison, accessible to the reader in the figure.

---

## Author Response (AR2)

December 28, 2023

Dear Dr. Evgeny Podolskiy,

Many thanks to the two reviewers and yourself (as Editor) for your positive appraisals. We're very pleased to read that this m/s is close to being accepted for publication in TC.

As it is important to carry out the unsupervised learning component (Part B, i.e. this contribution) on a dataset of cryoseismic signals that has broad scope, we'd prefer to retain the citations/pointers in the companion m/s (Part A). Hence, we will proceed according to your suggestion to resubmit final files and await your decision on Part A.

Regarding Part A, we're pleased to read that one reviewer now recommends that it (Part A) is suitable for publication. The remaining concerns of the other reviewer relate to disconnects on points that we are happy to clarify and/or provide additional information on.

Best,
Rebecca Latto